# Comparison of Air Quality at Different Altitudes from Multi-Platform Measurements in Beijing

Hongzhu Ji[1], Siying Chen[1, *], Yinchao Zhang[1], He Chen[1], Pan Guo[1], Peitao Zhao[2]

[1]School of Optics and Photonics, Beijing Institute of Technology, Beijing 100081, China
[2]China Meteorological Administration, Beijing 100081, China

*Correspondence to*: Siying Chen (csy@bit.edu.cn)

**Abstract.** The features of upper-air visibility at altitudes of 0.1km, 0.3km and 0.5km and the two-dimensional haze characteristics in the northwest of downtown Beijing were studied by using a multiplatform analysis during haze episodes between December 17th, 2016 and January 6th, 2017. Through the multiplatform data analysis in hourly and daily variation, the upper-air visibility increased along with the decrease of PM2.5 mass concentration. And the upper-air visibility on non-haze days was about 3-5 times higher than that on haze days. The vertical transport of pollutants can be inferred from the delayed variation of upper-air visibility between high altitude and low altitude. In addition, the two-dimensional haze characteristics could be studied by analyzing the correlation between vertical haze parameters (atmospheric boundary layer, haze thickness and aerosol optical thickness) and horizontal haze parameter (upper-air visibility). The characteristics of multi-parameters have been analyzed and concluded for different haze levels. Such analyses are useful to understanding the air pollution transport, as well as uncertainties in using ground-based measurements to represent column values.

## 1 Introduction

Due to increasing anthropogenic emissions resulting from China's rapid economy growth and urbanization, haze pollution has been a common problem in East Asia, especially China (Han et al., 2016; Guan et al., 2017; Liu et al., 2013). During the past two decades, scientists have carried out many experiments to explain the formation and evolution mechanism of haze (Chen and Wang, 2015; Tao et al., 2014; Wu et al., 2012; Xin et al., 2014; Zhao et al., 2017). According to research of Chen and Wang, the annual haze days in North China were relatively few in the 1960s, but increased sharply in the 1970s and have remained stable to the present through the analysis of long-term variation during the period of 1960-2012 (Chen and Wang, 2015). To characterize the haze phenomena, it is important to understand the haze parameters determined by aerosol optical properties. It is known that visibility mainly reflects the information of horizontal extinction near the surface and can be considered a good indicator of haze pollution (Sun et al., 2016; Yang et al., 2013). According to research of Wu et al. (2012), the visibility on sunny days in 543 stations in China were analyzed and the results indicated the annual mean visibility on sunny days is higher in Northwest China and lower in Southeast China, which is similar to the distribution of

aerosol optical thickness (AOT). The visibility impairment is attributed to the scattering and absorption of the particulate and gaseous pollutants in the atmosphere (Mishra and Kulshrestha, 2016; Song et al., 2003; Yang et al., 2007).

The height of the atmospheric boundary layer (ABL) is an important parameter to study the remote sensing of particulate matters near the ground, which has the closest relationship with human activities and the ecological environment (Amiridis et al., 2007; Dong et al., 2017; Sawyer and Li, 2013; Stull, 2012). It could also adjust due to surface effect within one hour (Chen et al., 2016; Wu et al., 2013; Zhang et al., 2013). Satellite observations show that haze pollution forms widespread haze layers like low clouds over China, which are usually called "haze clouds" to indicate their large coverage (Tao et al., 2012; Tao et al., 2014). The dense haze layer can evidently alter regional radiation and the hydrological cycle. Then the near-surface horizontal visibility will be further impaired due to radiative feedback (Gao et al., 2015; Qian et al., 2009). Tao et al. (2014) presented the formation and variation of thick haze layers are mostly associated with regional transport and moist airflows. AOT is defined as the extinction of monochromatic light due to the presence of aerosols in the atmosphere, and can be retrieved by the integration of aerosol extinction coefficient over a certain vertical distance. Many researchers have reported the importance of AOT to visibility (Alexandrov et al., 2016; B äumer et al., 2008; Dong et al., 2017; Li et al., 2007; Xin et al., 2014). Through observing the deterioration process of air quality in Germany, B äumer et al. (2008) found that a distinct decreasing trend in visibility was accompanied by a significant increase in AOT. So far, many researches have been conducted to study the effect of different haze parameters on visibility. However, the above research mainly focused on the horizontal visibility near the ground, and less focus was attached to the characteristics of upper-air visibility (Up-Vis). Moreover, the research has been hardly found to report the two-dimensional haze characteristics. Therefore, further studies are necessary to analyze these characteristics.

In this paper, the characteristic of Up-Vis and potential correlation with various vertical haze parameters (ABL, AOT and haze thickness) were investigated in the northwest of downtown Beijing during haze episodes between December 17th, 2016 and January 6th, 2017. The research was conducted by using the ground-based Raman-Mie LiDAR, meteorological ground-based observation equipment, and the ground-based remote sensing aerosol robotic network (AERONET). This paper aims to (1) present the hourly and daily variation of haze parameters during haze episode in the northwest of downtown Beijing; (2) reveal the impact of the vertical transport of PM2.5 (particulate matter with a diameter less than 2.5 μm) mass concentration on Up-Vis and investigate the two-dimensional haze phenomenon based on the correlation between vertical haze parameter (ABL, AOT and haze thickness) and horizontal haze parameter (Up-Vis); (3) understand the classification standard of haze levels, proposed by World Meteorological Organization (WMO), based on the multi-parameter analysis.

## 2 Methodology

### 2.1 Sites description

Figure 1 shows the geographic coordinates of multiplatform sites, including one ground-based LiDAR detecting site (denoted as star), three air quality monitoring sites (denoted as circle) and four AERONET sites (denoted as upper triangle).

The ground-based Raman-Mie LiDAR site is located at the LiDAR Lab of Beijing Institute of Technology in Beijing, China. The detected pure rotational Raman and elastic returns are used for obtaining the vertical characteristic of aerosols. The selected three air quality monitoring sites around the LiDAR site include Xizhimen north site, Wanliu and Guanyuan site. The PM2.5 mass concentration is one of the variables to be monitored. The data collected from four AERONET sites, including Beijing site, Beijing_RADI site, Beijing_PKU site and Beijing_CAMS site, are used to acquire the AOT value in LiDAR site by using statistical calculation. The distances between the LiDAR site and other sites range from 2.63km to 7.59km. Moreover, the periods of all the downloading PM2.5 mass concentration data and AOT data are the same as the detecting time, between December 13th, 2016 and January 11th, 2017, of ground-based LiDAR site.

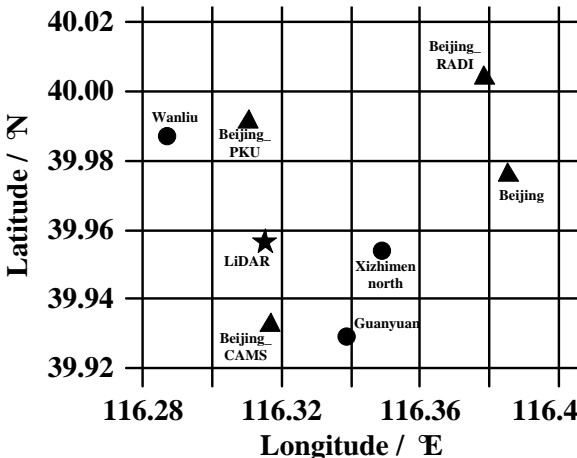

Figure 1: Geographic coordinate of ground-based LiDAR site (star), air quality monitoring sites (circle) and AERONET sites (upper triangle).

## 2.2 Data analysis method

To obtain the PM and AOT values in the ground-based LiDAR site accurately and reliably, three air quality sites and four AERONET sites in Fig. 1 are selected for collecting data. According to the distance information between the LiDAR site and the selected sites, the PM and AOT values at the LiDAR site are calculated with the following statistical equation:

$$PM = \sum_{i=1}^{n} G_i PM_i \quad (\sum_{i=1}^{n} G_i = 1, \ n = 3), \tag{1}$$

$$AOT = \sum_{i=1}^{m} Q_i AOT_i \quad (\sum_{i=1}^{m} Q_i = 1, \ m = 4), \tag{2}$$

$PM_i$ represents the PM value of the selected three air quality sites supplied by the Beijing Municipal Environmental Monitoring Centre (BJMEMC); $AOT_i$ describes the AOT value of the four AERONET sites; $G_i$ and $Q_i$ denote the normalized weight function which is inversely proportional to the distance between LiDAR site and the selected sites. According to the definition of AOT, it can be obtained by the integration of aerosol extinction coefficient over a certain vertical distance with the expression of $\int_0^z \alpha_a(z)dz'$, where $\alpha_a(z)$ is the aerosol extinction coefficient (AEC) which is retrieved from ground-based Raman-Mie LiDAR data with some robust inversion methods (Ji et al., 2017). As shown in Fig. 2, it is believed that the

AOT value deduced from ground-based LiDAR data is reasonable and reliable due to the excellent Pearson correlation coefficient (+0.87) and $R^2$ value (0.75). Besides, AOT is classified as vertical haze parameter because of its representative significance to pollutant concentration at a certain vertical distance.

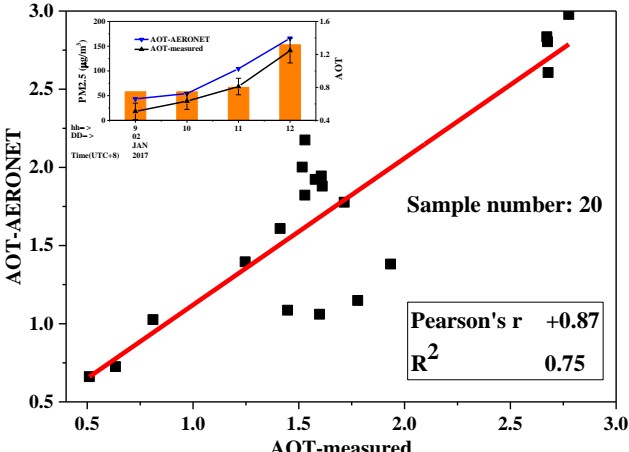

5 **Figure 2: Correlation of the AOT values deduced from AERONET sites and ground-based LiDAR data. The inserted chart gives the changes of PM2.5 mass concentration and AOT values at the ground-based LiDAR site on January 2, 2017.**

The Up-Vis is defined as the horizontal visibility at different altitudes, which is classified as horizontal haze parameter. According to the Koschmieder's formula (Larson and Cass, 1989; Lee and Shang, 2016), the Up-Vis at a certain altitude is calculated with the following equation:

$$V(z) = -\ln A \big/ b_{ext}(z),$$ (3)

where $A$ is the limiting contrast threshold for the average human observer, with the common value of 0.02 (Middleton, 1957). $b_{ext}$ is the total extinction coefficient. According to the research of Song et al. (2003), the visibility impairment mainly depends on the light scattering extinction by particles in $b_{ext}$.

According to the observation and forecasting level of haze (QX/T 113-2010) and the requirements for human health (Jarraud, 15 2008; Han et al., 2016), when the Up-Vis at a certain altitude is about 5km based on Eq. (3), the haze thickness (HT) can be defined as the value of this altitude. Therefore, HT reflects the main region of high concentration pollutions and can be classified as vertical haze parameter.

The height of ABL is affected by the underlying surface, and can be retrieved by detecting the rapid drop-off in extinction or backscatter coefficient between the free troposphere and the mixing layer as shown in the following equation (Flamant et al., 20 1997; Sawyer and Li, 2013):

$$h_{ABL} = \max \left| \frac{\partial \alpha_a(z)}{\partial z} \right|,$$ (4)

Tang et al. (2015) indicated the ABL represents the atmospheric diffusion capacity in vertical direction, so it can be classified as the vertical haze parameter.

## 3 Results and discussion

Figure 3 shows the space-time diagram of AEC by analyzing the detected data from ground-based LiDAR during two successive haze episodes in the northwest of downtown Beijing. Figure 3a shows the height of the haze layer (denoted as high extinction area) increased to the maximum value at 5:00 p.m. on December 20, 2016, afterwards, the haze almost dissipates at 3:00 a.m. on December 22, 2016. A thicker haze layer of about 0.6km could be generally observed as shown in Fig. 3b. Moreover, the variation of some haze parameters would be further obtained by analyzing the two successive haze episodes, which is detailed in the sections below. Section 3.1 denotes the hourly changes of multiplatform data, section 3.2 shows the daily variation of multiplatform data, and section 3.3 presents the relationship between multiple parameters.

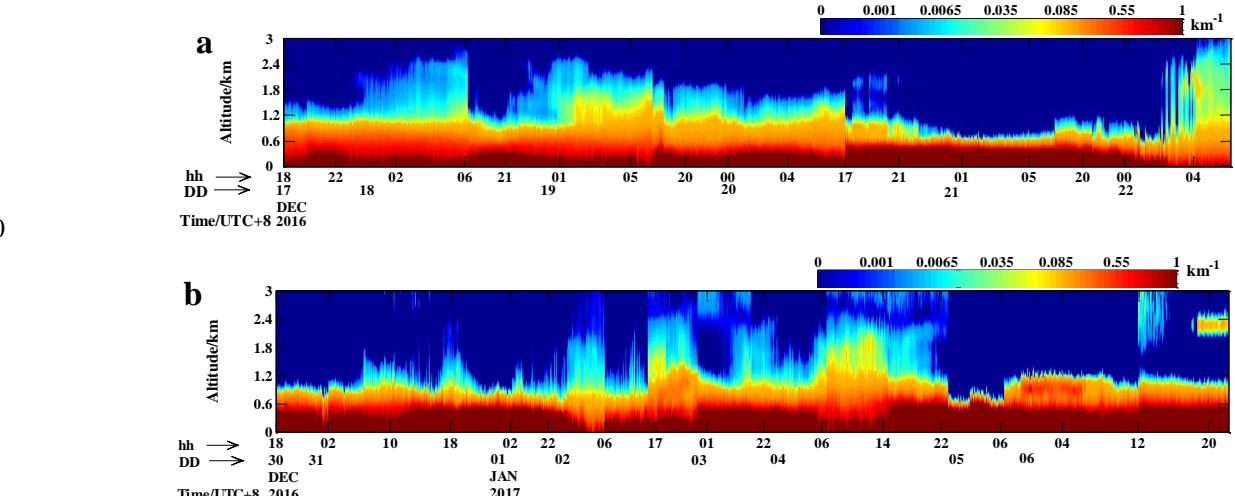

**Figure 3: Space-time diagram of AEC in the northwest of downtown Beijing during two haze episodes near 2017 New Year's Day.**

### 3.1 Hourly variation of multiplatform data

Figure 4 plots the hourly variation of haze parameters and meteorological elements during the first haze episode shown in Fig. 3a. The meteorological elements include PM2.5 mass concentration supplied by BJMEMC, RH, temperature, WD, and WS supplied by China Meteorological Administration (CMA). It is shown that the maximum Up-Vis (about 7.1 km, 12.4 km, and 15 km at the altitudes of 0.1 km, 0.3 km and 0.5 km, respectively) and the maximum ABL height (about 0.9 km) were obtained at 6:00 a.m. on December 22, 2016, where the variation trend is in contrast to the PM2.5 mass concentration. However, the peak and valley values of HT and AOT, respectively occurred at 9:00 p.m. on December 21, 2016 and at 6:00 a.m. on December 22, 2016, following the same trend as the PM2.5 mass concentration. Influenced by the effects of relative humidity (RH), the high RH enhanced the photochemical transformation of secondary aerosols that leads to a higher concentration of fine mode particles, which exacerbates the atmospheric elements, for example, impairment of Up-Vis, turbulence in ABL, and increase in HT and AOT (Hennigan et al., 2008). According to the topographic feature of Beijing,

the strong north wind can accelerate the diffusion of pollutants that gradually reduce the haze pollution after December 22, 2016. Moreover, the error bars indicate data uncertainty, which may originate from the fluctuation of signals by the natural variability of the atmosphere and the inaccurate calibration parameter of the inversion method.

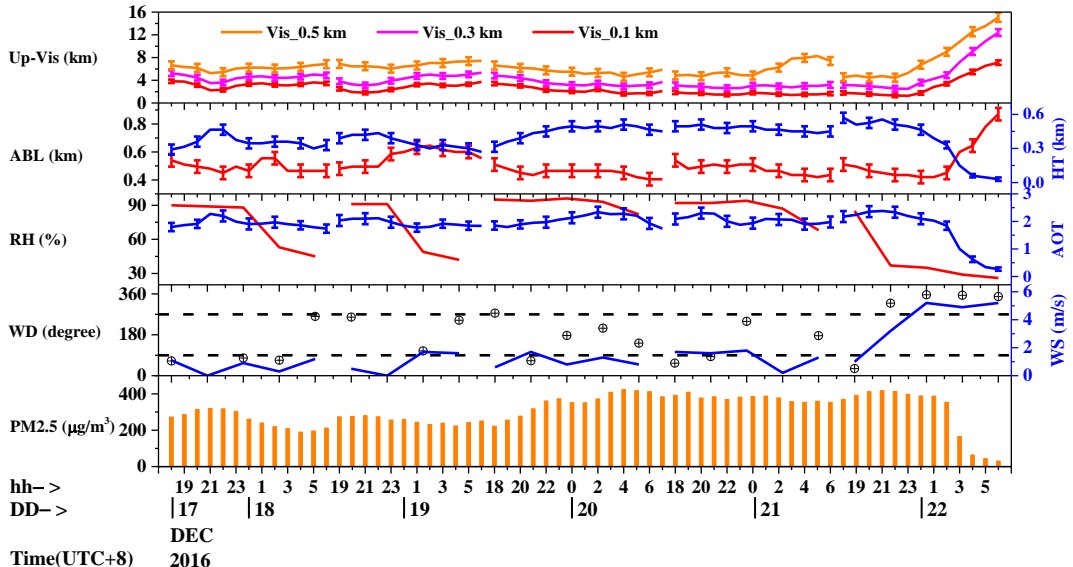

**Figure 4: Hourly variation of multiplatform data between December 17, 2016 and December 22, 2016 in the northwest of downtown Beijing.**

Similar results can be found in the other haze episodes shown in Fig. 5. The Up-Vis and ABL have a negative correlation with the tendency of PM2.5 mass concentration. The Up-Vis reached peak values of about 5 km, 9.3 km, and 13.6 km at the altitudes of 0.1 km, 0.3 km, and 0.5 km, respectively in the daytime on January 2, 2017, where the Up-Vis corresponded to the PM2.5 mass concentration of 58μg/m$^3$ and a smaller RH of 55%. On the contrary, the maximum HT and AOT of about 0.8 km and about 3.6 was detected at 2:00 a.m. on January 4, 2017, which corresponded to the PM2.5 mass concentration of 561μg/m$^3$ and a larger RH of 97%. In addition, the continuous moderate pollution after January 5, 2017 could be attributed to the strong north wind with a maximum wind speed of 3 m/s in the night time of January 4, 2017 and the weak south wind with a mean wind speed of about 1.3 m/s on January 6, 2017 (Han et al., 2016; Zhao et al., 2013). A higher PM2.5 mass concentration led to the increase in AOT, which was accompanied by the decrease in Up-Vis, as derived by Dong et al. (2017) from a combination of the Moderate-Resolution Imaging Spectroradiometer (MODIS) and the Multi-angle Imaging SpectroRadiometer (MISR) across Guanzhong Plain. Additionally, the error bars indicate data uncertainty, which may originate from the fluctuation of signals caused by the natural variability of the atmosphere and the inaccurate calibration parameter of the inversion method.

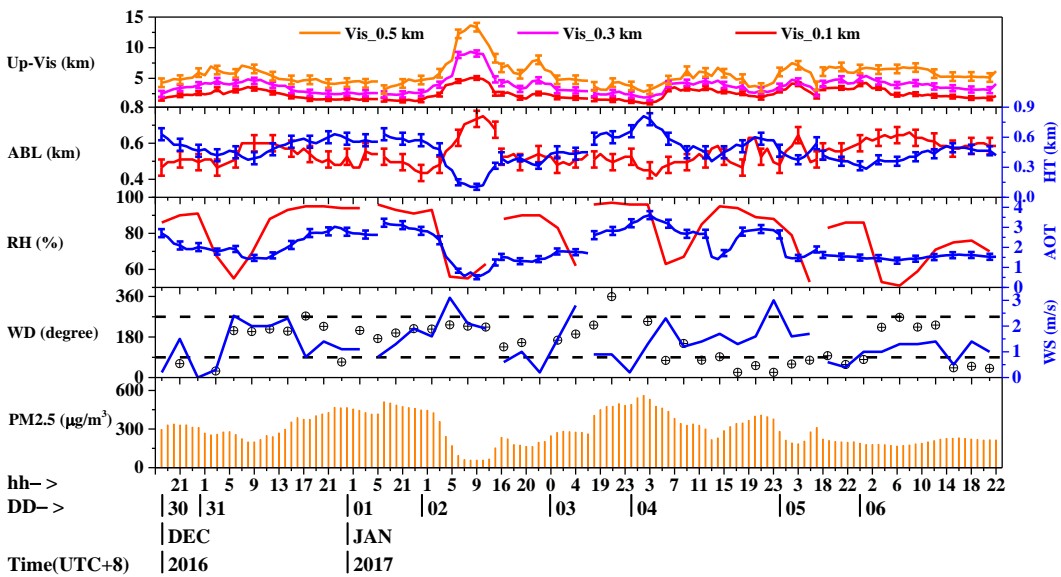

**Figure 5: Hourly variation of multiplatform data between December 30, 2016 and January 6, 2017 in the northwest of downtown Beijing.**

## 3.2 Daily variation of multiplatform data

5    To compare and analyze the difference of haze parameters on haze days and non-haze days, Fig. 6 presents the daily variation of Up-Vis, ABL, HT, and AOT with the meteorological elements. The haze days are shown in the areas highlighted in grey in Fig. 6. The following phenomena are concluded from Fig. 6: (1) the minimum Up-Vis values were about 1.5 km, 2.5 km, and 4.2 km at the altitudes of 0.1 km, 0.3 km, and 0.5 km, respectively. The Up-Vis on non-haze days was about 3-5 times higher than that on haze days. (2) The height of ABL was about 0.5 km on haze days, and ranged from 0.6 km to 0.9

10  km on non-haze days. (3) The trends that contradicted to the Up-Vis and ABL could be found in the results of HT and AOT. By combining meteorological elements, a lower Up-Vis and higher HT can be measured when PM2.5 and RH values were higher and the wind blew from the south. Moreover, when the prevailing wind came from the north and the RH value decreased, the diffusion of pollutants was accelerated, which improved the air quality and enhanced the Up-Vis. A high RH may favour the local contribution of humidity-related physicochemical processing in haze pollution, so the Up-Vis decreased

15  on haze days, which is similar to the research from Tang et al. (2015). In addition, the error bars indicate data uncertainty, which may originate from the fluctuation of signals caused by the natural variability of the atmosphere and the inaccurate calibration parameter of the inversion method.

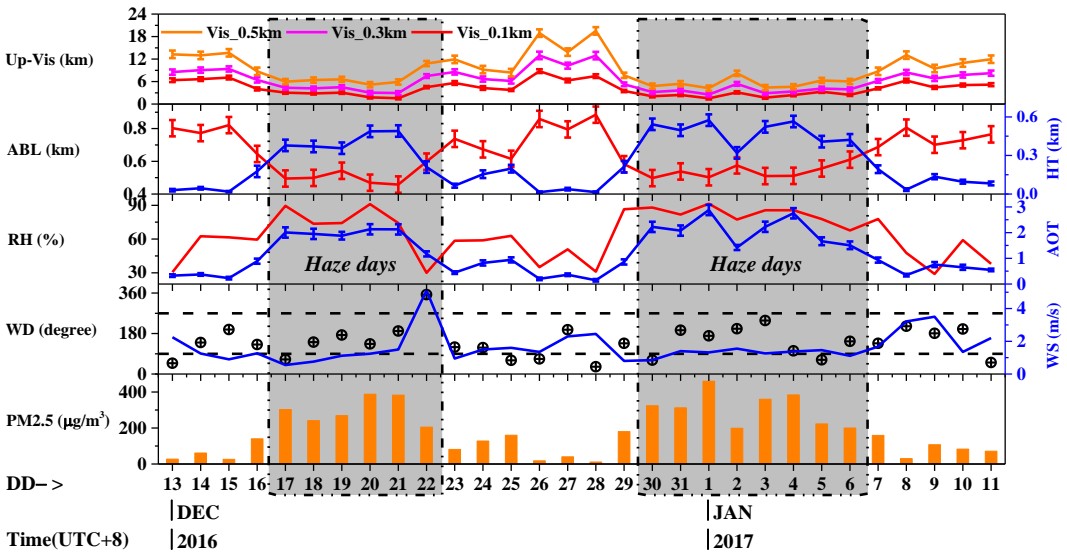

**Figure 6: Daily variation of multiplatform data during successive haze episodes in the northwest of downtown Beijing.**

### 3.3 Correlation between Up-Vis, ABL, HT, AOT, and PM2.5 mass concentration

As shown in Fig. 7, the correlation between PM2.5 mass concentration and haze parameters was established based on the 201 statistical samples in Fig. 4 and 5, which describes the impact of near-ground particle concentration on haze parameters in the northwest of downtown Beijing. Figure 7a and 7b plot the exponential reduction of the ABL and Up-Vis values when PM2.5 mass concentration increased, with $R^2$ values at about 0.73 (mean value of 0.76, 0.81, and 0.62) and 0.62, respectively. Moreover, owing to the location of detecting sites (located at the centre of Beijing) and the different influence of human activities on Up-Vis at individual altitudes, the correlations between surface PM2.5 and Up-Vis at altitudes of 0.3 km and 0.5 km (0.81 and 0.76 respectively) are much stronger than the correlation between surface PM2.5 and Up-Vis at altitude of 0.1 km (0.62). In Fig. 7a, with the decreasing of PM2.5 mass concentration, the Up-Vis at the altitude of 0.1 km gradually increases, but the Up-Vis at the altitudes of 0.3 km and 0.5 km increases much faster as shown in the inserted table. The exponential correlation between ABL height and PM2.5 mass concentration is similar to the studies of Zhao et al. (2017). From Fig. 7c and 7d, it can be observed that the HT and AOT values increased linearly with the growing PM2.5 mass concentration, with the $R^2$ values at 0.75 and 0.84, respectively. With the accumulation of pollutants, the aerosol column concentration and the PM2.5 mass concentration would increase, which aggravates the light scattering and absorption. Therefore, the near-ground fine pollutant concentration has a significant influence on haze parameters, so the haze could be alleviated by controlling fine pollutant concentrations near the ground.

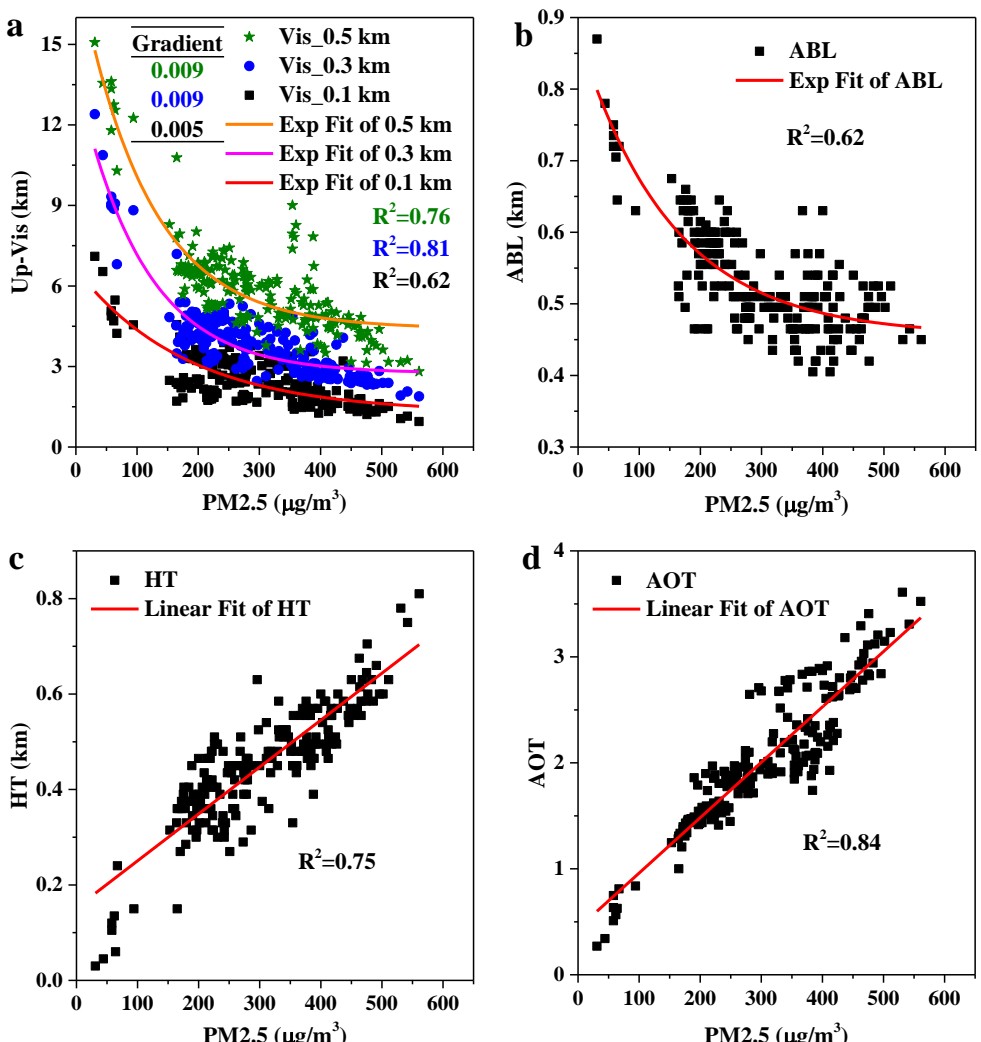

**Figure 7: Scatter plot of PM2.5 mass concentration and haze parameters of Up-Vis, ABL, HT, and AOT in the northwest of downtown Beijing. The inserted table in Fig. 7a denotes the statistical gradient of Up-Vis at different altitudes.**

5    As shown in Fig. 8, the vertical transport of particles could be obtained by comparing hourly variations of PM2.5 mass concentration and Up-Vis at different altitudes in certain period. In Fig. 8 (1), as the PM2.5 mass concentration near the ground decreased, the Up-Vis at the altitude of 0.5 km increased three hours later than that at the altitudes of 0.1 km and 03 km. This indicates pollutants might ascend and prevents the improvement of Up-Vis at the altitude of 0.5 km. In Fig. 8 (2), the Up-Vis at the altitude of 0.5 km increased rapidly, while the Up-Vis at the altitudes of 0.1 km and 0.3 km increased

10   slowly four hours later. This demonstrates the delayed diffusion might result from the descent of pollutants. While the descent of pollutants cause that near-ground PM2.5 mass concentration decreased slowly in this period. Therefore, the delayed variations of Up-Vis between high altitude and low altitude indirectly reveal the influence of vertical transport of pollutants on variation of haze parameters.

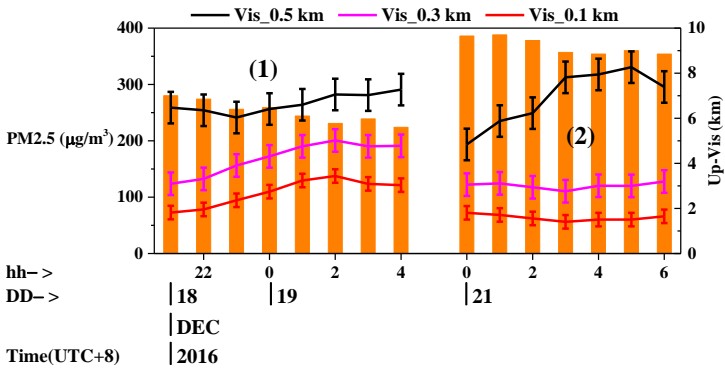

**Figure 8: Hourly variation of Up-Vis and PM2.5 mass concentration in certain period.**

According to the 201 statistical samples mentioned above, the correlations between vertical haze parameters (ABL, HT and AOT) and horizontal haze parameters (Up-Vis) are plotted in Fig. 9 to analyze the two-dimensional characteristic of haze

5   phenomenon. Figure 9a shows a positive exponential correlation between ABL and Up-Vis, with $R^2$ values of 0.44, 0.58, and 0.46 at the altitudes of 0.1 km, 0.3 km, and 0.5km, respectively. Tang et al. (2015) indicated the ABL can represent the atmospheric diffusion capacity in vertical direction, so the increase in Up-Vis was accompanied by the increase in ABL. However, when the HT or AOT values increased, the Up-Vis would decrease exponentially as shown in Fig. 9b and 9c. Compared with the studies of Dong et al. (2017), the similar anticorrelation can be inferred between visibility and AOT. And

10   the exponential changes in Up-Vis and AOT or HT could be attributed to the rapid accumulation of aerosol particles near the surface. The table 1 shows the statistical gradient of Up-Vis at different altitudes changing with the vertical haze parameters. It is obvious that the Up-Vis at altitude of 0.3 km changed faster than that at altitudes of 0.1 km and 0.5 km. Therefore, through the analysis of the correlation between vertical haze parameters (ABL, HT and AOT) and horizontal haze parameter (Up-Vis), the haze characteristics could be well investigated in two dimensions.

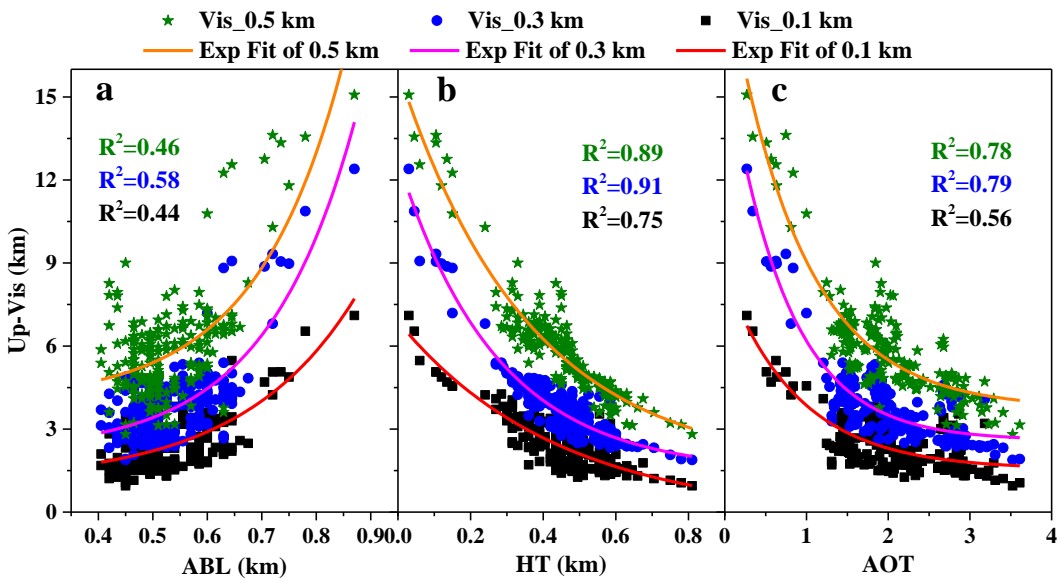

**Figure 9: Scatter plot of Up-Vis and vertical haze parameters of ABL, HT, and AOT in the northwest of downtown Beijing.**

**Table 1: Statistical gradient of Up-Vis with different vertical haze parameters at different altitudes for Fig. 9.**

| Vertical haze parameters | Vis_0.1 km | Vis_0.3 km | Vis_0.5 km |
|---|---|---|---|
| ABL | 4.801 | 6.246 | 6.101 |
| HT | 2.275 | 3.674 | 2.787 |
| AOT | 1.108 | 1.365 | 1.111 |

According to the observation and forecasting levels of haze (QX/T 113-2010) supplied by CMA, there are four forecasting levels of haze: slight pollution, mild pollution, moderate pollution and severe pollution (CMA, 2010). Table 2 provides the standard range of horizontal visibility on the surface (H-Vis) for different haze levels. When slight pollution occurred with the H-Vis of 5-10 km, the corresponding PM2.5 mass concentration is less than $60\pm20\mu g/m^3$, the Up-Vis at the altitudes of 0.1 km, 0.3 km, and 0.5 km is larger than $6.5\pm0.3$ km, $9\pm0.5$ km, and $14\pm1$ km, respectively, and the ABL is higher than $0.8\pm0.03$ km. When mild pollution occurred with the H-Vis of 3-5 km, the minimum Up-Vis decreased to $3.8\pm0.2$ km, $5.1\pm0.2$ km, and $7.2\pm0.3$ km at the altitudes of 0.1 km, 0.3 km, and 0.5 km, respectively. While the ABL would also decline, with the minimum value of $0.57\pm0.03$ km. However, the AOT would increase from $0.4\pm0.05$ to $1.5\pm0.1$. When the H-Vis value is between 2 km and 3 km, the haze level is classified as moderate pollution. The PM2.5 mass concentration changes from $150\pm30\mu g/m^3$ to $300\pm40\mu g/m^3$. The Up-Vis would decrease from the minimum value of mild pollution to $2.6\pm0.1$ km, $3.7\pm0.2$ km, and $5.2\pm0.2$ km at the altitudes of 0.1 km, 0.3 km, and 0.5 km, respectively. Simultaneously, the HT of between $0.3\pm0.03$ km and $0.48\pm0.03$ km could be obtained. Once the H-Vis is lower than 2 km, severe pollution would occur, with the corresponding PM2.5 mass concentration higher than $300\pm40\mu g/m^3$. The Up-Vis would decrease further based on the minimum value of moderate pollution, and the turbulent ABL height could range from $0.42\pm0.03$ km to $0.5\pm0.03$ km.

Moreover, the HT and AOT would further increase. Therefore, the obtained variation range of different haze parameters would be helpful for understanding the atmospheric elements of different haze levels through multi-parameter analysis and for serving the haze governance.

**Table 2: Values of Haze parameters and meteorological elements corresponding to the haze levels.**

| Parameters | | Slight pollution | Mild pollution | Moderate pollution | Severe pollution |
|---|---|---|---|---|---|
| H-Vis (km) | | 5-10 | 3-5 | 2-3 | <2 |
| PM2.5 (μg/m³) | | <60±20 | 60±20-150±30 | 150±30-300±40 | >300±40 |
| *Up-Vis (km)* | *0.1 km* | *>6.5±0.3* | *3.8±0.2-6.5±0.3* | *2.6±0.1-3.8±0.2* | *<2.6±0.1* |
| | *0.3 km* | *>9±0.5* | *5.1±0.2-9±0.5* | *3.7±0.2-5.1±0.2* | *<3.7±0.2* |
| | *0.5 km* | *>14±1* | *7.2±0.3-14±1* | *5.2±0.2-7.2±0.3* | *<5.2±0.2* |
| ABL (km) | | 0.8±0.03-1[a] | 0.57±0.03-0.8±0.03 | 0.5±0.03-0.57±0.03 | 0.42±0.03-0.5±0.03 |
| HT (km) | | ≈0 | <0.3±0.03 | 0.3±0.03-0.48±0.03 | >0.48±0.03 |
| AOT | | <0.4±0.05 | 0.4±0.05-1.5±0.1 | 1.5±0.1-2.1±0.2 | >2.1±0.2 |

[a]: Range of ABL is 0.3-1 km (Garratt, 1994).

## 4 Conclusions

In this study, the traits of upper-air visibility and the two-dimensional haze characteristic were investigated during the haze episodes between December 17th, 2016 and January 6th, 2017 in the northwest of downtown Beijing by using a multiplatform analysis. The close connection with AERONET's statistical results demonstrates that the retrieved aerosol extinction coefficient is reliable and believable. And compared with the changes of PM2.5 mass concentration, an opposite tendency can be found for Up-Vis by hourly and daily haze analysis. The Up-Vis on non-haze days was about 3-5 times higher than that on haze days with the ground-based Raman-Mie LiDAR data between December 13th, 2016 and January 11th, 2017. The higher relative humidity would aggravate the haze characteristics owing to the enhanced photochemical transformation of secondary aerosols. On the contrary, the strong north wind would accelerate the diffusion of pollutants due to the topographic feature of Beijing. Moreover, a strong correlation between PM2.5 mass concentration and haze parameters shows an obvious influence of near-ground fine particle concentration on haze parameters, so the haze phenomenon could be alleviated by controlling fine pollutant concentrations near the ground. In addition, the delayed variations of Up-Vis between high altitude and low altitude reveal the vertical transport of pollutants. The correlation between vertical haze parameters (ABL, AOT and HT) and horizontal haze parameter (Up-Vis) can help investigate the two-dimensional characteristics of haze phenomenon. Besides, the proposed variation range for different haze parameters would be beneficial to understand the atmospheric conditions of different haze levels through multi-parameter analysis and to serve the haze governance.

*Acknowledgments*. This research is supported by the National Natural Science Foundation of China (No. 61505009).

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
