# Peer review of "Multiplatform analysis of upper-air haze visibility in downtown Beijing"

_Atmospheric Chemistry and Physics, 2018_

## Referee Comment (RC1) · Anonymous Referee #1 · 22 Mar 2018

I can't find the scientific significance and academic value of this article. Why study Up-Vis ïijĹ0.1,0.3,0.5kmïijĽïij§The authors have not pointed out the differences of Up-Vis at the three altitudes, nor have they studied the differences between them and the horizontal visibility on the ground. What is the purpose of using Up-Vis at three high altitudes? There are many detailed studies focusing on the relationship between visibility, relative humidity and other meteorological elements and PM2.5. The study method and conclusions of this article are too simple and general comparing with the related works.

---

## Referee Comment (RC2) · Anonymous Referee #2 · 26 Mar 2018

Major comments:

The scientific significance of this study is not clear enough to me. Why the authors investigate the relationship between haze parameters and upper air visibility? How important of the upper air visibility and the results of this study on the understanding of haze phenomena? Thus, I'd suggest giving more description on this.

The data analysis and discussion are very shallow and on the surface, and thus more studies and deep discussions should be made to make the study original enough. Furthermore, I was left wondering to what new understanding we are able to take away from the study.

Minor comments:

[Figure]

P1, Abstract: it is better to give the periods of the haze episodes, and which altitude of the upper air and what the haze parameters refer to here.

P1, L19: "the haze days have shown a marked increase in years before 2006." How about the years after 2006? Is it after 2006 here?

P3, 2.2: please give the short description on how to get the AOD from lidar measurements, how about the uncertainties?

P4, figure 2: Only one day's data is used to validate the lidar retrieved AOD, is it because only one day retrieval available?

P4, figure3: The AEC determined from lidar is only for cloud-free conditions or all conditions?

P5, L3: "the haze parameters would alter with the hourly and daily changes of haze level", this result is well known. The figures 3 just give the variation of haze height.

P7, L7: on haze day or non-haze day?

P8, L11: I do not see the results of "the spatial transport of pollutants has a significant effect on haze parameters" can be concluded from the above description.

Figure 7: please give the number of samples.

P10, L5: please give the standard of the four haze levels

---

## Author Comment (AC1) · 27 Apr 2018

We deeply appreciate the reviewer for his/her careful reviews of this paper.

As we've known, the haze thickness (HT) is defined as the altitude where the upper air visibility (Up-Vis) is about 5 km (Han et al., 2016). This demonstrates the parameter of Up-Vis is key for obtaining the variation of HT. It is shown in Fig. 6 that the HT changes from 0.3 km to 0.6 km on haze days. So the Up-Vis at the altitudes of 0.1 km, 0.3 km, and 0.5 km are studied to characterize the HT. In addition, the Up-Vis at other altitudes can also be selected to study the characteristics of HT, as shown in Fig. R1. In Fig. R1, the similar phenomenon can be found compared with the results of Fig. 6. But considering the standard appearance of graphics, only three typical altitudes are

shown in the paper.

According to the research of Tang et al. (2015), the atmospheric boundary layer (ABL) represents the atmospheric diffusion capacity in vertical direction, the aerosol optical thickness (AOT) directly reflects the particle concentration at a certain vertical distance, and the HT represents the main region of high concentration particles. The Up-Vis, the horizontal visibility at different altitudes, represents the horizontal particle concentration at a certain altitude. Therefore, the Up-Vis characterizes the horizontal haze situations at different altitudes; the ABL, AOT and HT characterize the vertical haze situation from different perspectives. And the correlation between vertical haze parameter (ABL, AOT and HT) and horizontal haze parameter (Up-Vis) characterizes the two-dimensional haze situations. Through comparing hourly variations of PM2.5 mass concentration and Up-Vis at different altitudes in certain period, the influence of vertical transport of pollutants on variation of haze parameters could be revealed indirectly. And according to the variation characteristics of Up-Vis and its correlation with vertical haze parameters (ABL, AOT and HT), the haze phenomenon in two dimensions can be analyzed, which provides more insights into haze phenomenon.

From Figs. 4-6, it is shown that the Up-Vis at the three altitudes have different variation ranges. The Figs. 7a shows the different correlation between the Up-Vis at the three altitudes and PM2.5 mass concentration. Figure 8 indicates the impact of vertical transport of pollutants on variation of haze parameters by analyzing the delayed variations of Up-Vis between high altitude and low altitude. Figure 9 reveals the correlation between horizontal haze parameter (Up-Vis at the three altitudes) and vertical haze parameters (ABL, AOT and HT). And Table 1 shows the statistical gradient of Up-Vis at different altitudes changing with the vertical haze parameters. Moreover, Table 2 displays the variation of Up-Vis at the three altitudes under different haze levels. Besides, the paper also indicates the minimum values of Up-Vis at the three altitudes are about 1.5 km, 2.5 km, and 4.2 km respectively on haze days, as shown in Lines 7-8 on Page 7. Therefore, the paper not only shows the numerical differences in Up-Vis at the

three altitudes qualitatively and quantitatively, but also shows the different correlation between Up-Vis and vertical haze parameters (ABL, AOT and HT).

Atmospheric visibility basically includes horizontal visibility, slant range visibility and vertical visibility (Hey, 2015). The upper air visibility (Up-Vis) is defined as the horizontal visibility at different altitudes which is detailed in Page 4. The Up-Vis at different altitudes is regarded as the horizontal visibility above the ground, and the horizontal visibility usually indicates the horizontal visibility near the ground. Until now, many studies on the visibility and its correlation with meteorological elements have been carried out to indicate the importance of visibility to air pollution studies (Yang et al., 2013; Sun et al., 2016; Wu et al., 2012; Bäumer et al., 2008; Pantazis et al., 2017). But these researches focus on the horizontal visibility and the slant range visibility rather than the upper air visibility. According to the obtained variation characteristics of Up-Vis, the influence mechanism of meteorological parameters to Up-Vis, and its correlation with vertical haze parameters (ABL, AOT and HT), the variation of Up-Vis would be significant to obtain the variation of haze thickness, and the haze phenomenon in two dimensions could be recognized, which provides more insights into haze phenomenon.

To be more scientific, we have changed the sentence "However, less focus was attached to the characteristics of upper air visibility (Up-Vis)." into "However, the above research mainly focused on the horizontal visibility near the ground, and less focus was attached to the characteristics of upper-air visibility (Up-Vis). Moreover, the research has been hardly found to report the two-dimensional haze characteristics." (see Lines 14-16 on Page 2).

To well demonstrate the two-dimensional haze characteristics, the sentence "In addition, a higher atmospheric boundary layer improves upper air visibility." has been changed into "In addition, the two-dimensional haze characteristics could be studied by analyzing the correlation between vertical haze parameters (atmospheric boundary layer, haze thickness and aerosol optical thickness) and horizontal haze parameter (upper-air visibility)." (see Lines 12-14 on Page 1). The sentence "(2) reveal the impact

of PM2.5 (particulate matter with a diameter less than 2.5 $\mu$m) mass concentration and haze parameters on upper air visibility;" has been changed into "(2) reveal the impact of the vertical transport of PM2.5 (particulate matter with a diameter less than 2.5 $\mu$m) mass concentration on Up-Vis and investigate the two-dimensional haze phenomenon based on the correlation between vertical haze parameter (ABL, AOT and haze thickness) and horizontal haze parameter (Up-Vis);" (see Lines 23-25 on Page 2). We have added the sentence "Besides, AOT is classified as vertical haze parameter because of its representative significance to pollutant concentration at a certain vertical distance." to classify the parameter of AOT (see Lines 19-20 on Page 3). The sentence "The Up-Vis is defined as the horizontal visibility at different altitudes." has been changed into "The Up-Vis is defined as the horizontal visibility at different altitudes, which is classified as horizontal haze parameter." (see Line 4 on Page 4). We have added the sentence "Therefore, HT reflects the main region of high concentration pollutions and can be classified as vertical haze parameter." to classify the parameter of HT (see Lines 13-14 on Page 4). The sentence "Tang et al. (2015) indicated the ABL represents the atmospheric diffusion capacity in vertical direction, so it can be classified as the vertical haze parameter." has been added to classify the parameter of ABL (see Lines 20-21 on Page 4). And the sentence "Therefore, a higher ABL has a positive influence on atmospheric visibility; and a lower HT or smaller AOT would enhance atmospheric visibility." has been changed into "The table 1 shows the statistical gradient of Up-Vis at different altitudes changing with the vertical haze parameters. It is obvious that the Up-Vis at altitude of 0.3 km changed faster than that at altitudes of 0.1 km and 0.5 km. Therefore, through the analysis of the correlation between vertical haze parameters (ABL, HT and AOT) and horizontal haze parameter (Up-Vis), the haze characteristics could be well investigated in two dimensions." (see Lines 11-14 on Page 10). The added table was shown in table R1 (see Table 1 on Page 11). We have changed the sentence "A higher ABL or lower HT as well as smaller AOT have a positive influence on the atmospheric visibility." into "The correlation between vertical haze parameters (ABL, AOT and HT) and horizontal haze parameter (Up-Vis) can help investigate the

**[ACPD](ACPD)**

Interactive
comment

[Figure]

two-dimensional characteristics of haze phenomenon." (see Lines 17-19 on Page 12).

Table R1: Statistical gradient of Up-Vis with different vertical haze parameters at different altitudes. Vertical haze parameters Vis_0.1 km Vis_0.3 km Vis_0.5 km ABL 4.801 6.246 6.101 HT 2.275 3.674 2.787 AOT 1.108 1.365 1.111

References: Bäumer, D., Vogel, B., Versick, S., Rinke, R., Möhler, O., and Schnaiter, M.: Relationship of visibility, aerosol optical thickness and aerosol size distribution in an ageing air mass over South-West Germany, Atmospheric Environment, 42, 989-998, 2008.

Han, R., Wang, S., Shen, W., Wang, J., Wu, K., Ren, Z., and Feng, M.: Spatial and temporal variation of haze in China from 1961 to 2012, Journal of Environmental Sciences, 46, 134-146, 2016.

Hey, J. D. V.: Determination of Cloud Base Height and Vertical Visibility from a Lidar Signal, Springer International Publishing, 2015.

Pantazis, A., Papayannis, A., and Georgousis, G.: Lidar algorithms for atmospheric slant range visibility, meteorological conditions detection, and atmospheric layering measurements, Applied Optics, 56, 6440, 2017.

Sun, T., Che, H., Wu, J., Wang, H., Wang, Y., and Zhang, X.: The variation in visibility and its relationship with surface wind speed in China from 1960 to 2009, Theoretical and Applied Climatology, 10.1007/s00704-016-1972-x, 2016.

Tang, G., Zhu, X., Hu, B., Xin, J., Wang, L., Münkel, C., Mao, G., and Wang, Y.: Impact of emission controls on air quality in Beijing during APEC 2014: lidar ceilometer observations, Atmospheric Chemistry and Physics, 15, 12667-12680, 2015.

Wu, J., Fu, C., Zhang, L., and Tang, J.: Trends of visibility on sunny days in China in the recent 50 years, Atmospheric Environment, 55, 339-346, 2012.

Yang, X., Ferrat, M., and Li, Z.: New evidence of orographic precipitation suppression

by aerosols in central China, Meteorology and Atmospheric Physics, 119, 17-29, 2013.

Please also note the supplement to this comment:
https://www.atmos-chem-phys-discuss.net/acp-2018-30/acp-2018-30-AC1-supplement.pdf

[Figure]

**Fig. 1.** Daily variation of upper air visibility during successive haze episodes in the northwest of downtown Beijing.

---

## Author Comment (AC2) · 27 Apr 2018

We deeply appreciate the reviewer for his/her careful reviews of this paper.

1. Major comments: The scientific significance of this study is not clear enough to me. Why the authors investigate the relationship between haze parameters and upper air visibility? How important of the upper air visibility and the results of this study on the understanding of haze phenomena? Thus, I'd suggest giving more description on this. The data analysis and discussion are very shallow and on the surface, and thus more studies and deep discussions should be made to make the study original enough. Furthermore, I was left wondering to what new understanding we are able to take away from the study.

[Figure]

Response:

The aerosol extinction coefficient can be retrieved from the ground-based LiDAR data, and is used to get the upper-air visibility (Up-Vis) at certain altitude, the aerosol optical thickness (AOT) at a certain vertical distance and the height of atmospheric boundary layer (ABL). As we've known, the haze thickness (HT) is defined as the altitude where the Up-Vis is about 5 km (Han et al., 2016). Moreover, according to the research of Tang et al. (2015), the ABL represents the atmospheric diffusion capacity in vertical direction, the AOT directly reflects the particle concentration at a certain vertical distance, and the HT represents the main region of high concentration particles. The Up-Vis, the horizontal visibility at different altitudes, represents the horizontal particle concentration at a certain altitude. Therefore, the Up-Vis characterizes the horizontal haze situations at different altitudes; the ABL, AOT and HT characterize the vertical haze situations from different perspectives. And the correlation between vertical haze parameter (ABL, AOT and HT) and horizontal haze parameter (Up-Vis) characterizes the two-dimensional haze situations.

Through comparing hourly variations of PM2.5 mass concentration and Up-Vis at different altitudes in a certain period, the influence of pollutants' vertical transport on variation of haze parameters could be revealed indirectly. And according to the variation characteristics of Up-Vis and its correlation with vertical haze parameters (ABL, AOT and HT), the haze phenomenon in two dimensions would be recognized, which provides more insights into haze phenomenon.

To be more scientific, the term "haze parameter", including ABL, AOT and HT, has changed into the term "vertical haze parameter (ABL, AOT and HT)". We have changed the sentence "However, less focus was attached to the characteristics of upper air visibility (Up-Vis)." into "However, the above research mainly focused on the horizontal visibility near the ground, and less focus was attached to the characteristics of upper-air visibility (Up-Vis). Moreover, the research has been hardly found to report the two-dimensional haze characteristics." (see Lines 14-16 on Page 2). And we have added

the sentence "The close connection with AERONET's statistical results demonstrates that the retrieved aerosol extinction coefficient is reliable and believable." to indicate the retrieved aerosol extinction coefficient is reliable (see Lines 9-10 on Page 12). Finally, to demonstrate the influence of meteorological elements on haze, the sentence "The higher relative humidity would aggravate the haze characteristics owing to the enhanced photochemical transformation of secondary aerosols. On the contrary, the strong north wind would accelerate the diffusion of pollutants due to the topographic feature of Beijing." has been added (see Lines 12-14 on Page 12).

To well demonstrate the two-dimensional haze characteristics, the sentence "In addition, a higher atmospheric boundary layer improves upper air visibility." has been changed into "In addition, the two-dimensional haze characteristics could be studied by analyzing the correlation between vertical haze parameters (atmospheric boundary layer, haze thickness and aerosol optical thickness) and horizontal haze parameter (upper-air visibility)." (see Lines 12-14 on Page 1). The sentence "(2) reveal the impact of PM2.5 (particulate matter with a diameter less than 2.5 $\mu$m) mass concentration and haze parameters on upper air visibility;" has been changed into "(2) reveal the impact of the vertical transport of PM2.5 (particulate matter with a diameter less than 2.5 $\mu$m) mass concentration on Up-Vis and investigate the two-dimensional haze phenomenon based on the correlation between vertical haze parameter (ABL, AOT and haze thickness) and horizontal haze parameter (Up-Vis);" (see Lines 23-25 on Page 2). We have added the sentence "Besides, AOT is classified as vertical haze parameter because of its representative significance to pollutant concentration at a certain vertical distance." to classify the parameter of AOT (see Lines 19-20 on Page 3). The sentence "The Up-Vis is defined as the horizontal visibility at different altitudes." has been changed into "The Up-Vis is defined as the horizontal visibility at different altitudes, which is classified as horizontal haze parameter." (see Line 4 on Page 4). We have added the sentence "Therefore, HT reflects the main region of high concentration pollutions and can be classified as vertical haze parameter." to classify the parameter of HT (see Lines 13-14 on Page 4). The sentence "Tang et al. (2015) indicated the ABL represents the

atmospheric diffusion capacity in vertical direction, so it can be classified as the vertical haze parameter." has been added to classify the parameter of ABL (see Lines 20-21 on Page 4). And the sentence "Therefore, a higher ABL has a positive influence on atmospheric visibility; and a lower HT or smaller AOT would enhance atmospheric visibility." has been changed into "The table 1 shows the statistical gradient of Up-Vis at different altitudes changing with the vertical haze parameters. It is obvious that the Up-Vis at altitude of 0.3 km changed faster than that at altitudes of 0.1 km and 0.5 km. Therefore, through the analysis of the correlation between vertical haze parameters (ABL, HT and AOT) and horizontal haze parameter (Up-Vis), the haze characteristics could be well investigated in two dimensions." (see Lines 11-14 on Page 10). The added table was shown in table R1 (see Table 1 on Page 11). We have changed the sentence "A higher ABL or lower HT as well as smaller AOT have a positive influence on the atmospheric visibility." into "The correlation between vertical haze parameters (ABL, AOT and HT) and horizontal haze parameter (Up-Vis) can help investigate the two-dimensional characteristics of haze phenomenon." (see Lines 17-19 on Page 12). Table R1: Statistical gradient of Up-Vis with different vertical haze parameters at different altitudes. Vertical haze parameters Vis_0.1 km Vis_0.3 km Vis_0.5 km ABL 4.801 6.246 6.101 HT 2.275 3.674 2.787 AOT 1.108 1.365 1.111

To describe the impact of near-ground particle concentration on haze parameters, we have added and changed some sentences as below: (1) added the sentence "In Fig. 7a, with the decreasing of PM2.5 mass concentration, the Up-Vis at the altitude of 0.1 km gradually increases, but the Up-Vis at the altitudes of 0.3 km and 0.5 km increases much faster as shown in the inserted table." (see Lines 8-9 on Page 8). (2) changed the sentence "Therefore, the spatial transport of pollutants has a significant effect on haze parameters." into "Therefore, the near-ground pollutant concentration has a significant influence on haze parameters, so the haze could be alleviated by controlling pollutant concentrations near the ground." (see Lines 14-15 on Page 8). (3) inserted the table about statistical gradient of Up-Vis at different altitudes as shown in Fig. R1 and added the sentence "The inserted table in Fig. 7a denotes the statistical gradient of Up-Vis at

different altitudes." (see Line 4 on Page 9). (4) changed the sentence "A strong correlation between PM2.5 mass concentration and haze parameters shows the effect of spatial transport of particles on haze parameters." into "Moreover, a strong correlation between PM2.5 mass concentration and haze parameters shows an obvious influence of near-ground particle concentration on haze parameters, so the haze phenomenon could be alleviated by controlling pollutant concentrations near the ground." (see Lines 14-16 on Page 12).

2. Minor comments: P1, Abstract: it is better to give the periods of the haze episodes, and which altitude of the upper air and what the haze parameters refer to here.

Response:

We have added the periods of the haze episodes, the altitude of the upper-air visibility and the detailed haze parameters in the Abstract. To be more scientific, the sentence of "The vertical features of upper air visibility in the northwest. . .near the 2017 New Year's Day" has changed into "The features of upper-air visibility at altitudes of 0.1km, 0.3km and 0.5km and the two-dimensional haze characteristics in the northwest of downtown Beijing were studied by using a multiplatform analysis during haze episodes between December 17th, 2016 and January 6th, 2017." (see Lines 7-9 on Page 1). And we have changed the sentence "The strong correlation between PM2.5 mass concentration and haze parameters shows the effect of spatial transport of particles on haze parameters." into "The vertical transport of pollutants can be inferred from the delayed variation of upper-air visibility between high altitude and low altitude." (see Lines 11-12 on Page 1).

P1, L19: "the haze days have shown a marked increase in years before 2006." How about the years after 2006? Is it after 2006 here?

Response:

We have changed the sentence "According to researches of Wu et al. (2010) and Gao

(2008), the annual average haze days have shown a marked increase in years before 2006 in China." into "According to research of Chen and Wang, the annual haze days in North China were relatively few in the 1960s, but increased sharply in the 1970s and have remained stable to the present through the analysis of long-term variation during the period of 1960-2012 (Chen and Wang, 2015)." to add the description about the variation of haze days after 2006 (see Lines 20-23 on Page 1).

P3, 2.2: please give the short description on how to get the AOD from lidar measurements, how about the uncertainties?

Response:

Aerosol optical thickness (AOT) can be defined as the extinction of monochromatic light due to the presence of aerosols in the atmosphere. Based on the lidar equation, the aerosol extinction coefficient was retrieved by some robust inversion methods. Then the AOT can be retrieved by the integration of aerosol extinction coefficient over a certain vertical distance. Owing to the signal-to-noise ratio difference caused by the natural variability of the atmosphere and the calibration and estimation errors caused by the robust inversion methods, the retrieved AOT would cause some errors according to the error propagation theory. To be more scientific, we have added the sentence "AOT is defined as the extinction of monochromatic light due to the presence of aerosols in the atmosphere, and can be retrieved by the integration of aerosol extinction coefficient over a certain vertical distance." to describe the AOT (see Lines 9-10 on Page 2).

P4, figure 2: Only one day's data is used to validate the lidar retrieved AOD, is it because only one day retrieval available?

Response:

Figure 2 shows the correlation of AOT values deduced from AERONET sites and ground-based LiDAR data. By comparing the data of the deduced AOTs, there are 20 sets of matching data, as shown in Fig. R2. To display how AOT values deduced

from AERONET sites and ground-based LiDAR data alter along with the changes of PM2.5 mass concentration, the results in one day are selected to demonstrate the variation, as shown in the inserted chart of Fig. R2. The inserted chart indicates the AOT values increase with the increasing of PM2.5 mass concentration through analyzing the data in this day. By combining all the matching data, a higher correlation between AOT values deduced from AERONET data and that retrieved from ground-based LiDAR data can be obtained, which demonstrates the AOT values (that is to say the aerosol extinction coefficient) retrieved from ground-based LiDAR data is reasonable and reliable. To be more scientific, we have added the number of matching samples into the figure (see Figure 2 on Page 4).

P4, figure3: The AEC determined from lidar is only for cloud-free conditions or all conditions?

Response:

The aerosol extinction coefficient (AEC) obtained from LiDAR data can be used for analyzing all conditions including the atmospheric characteristics below more than 10 km. Owing to the severe extinction caused by the existing haze, the detection altitude may not reach the position of cloud. Moreover, the haze mainly concentrates within about 1 km. Therefore, the plotted interval of Y-axis in Fig. 3 is below 3 km.

P5, L3: "the haze parameters would alter with the hourly and daily changes of haze level", this result is well known. The figures 3 just give the variation of haze height.

Response:

Thanks for your valuable comment. We have changed the sentence "Therefore, the haze parameters would alter with the hourly and daily changes of haze level, which will be described in details in the sections below." into "Moreover, the variation of some haze parameters would be further obtained by analyzing the two successive haze episodes, which is detailed in the sections below." (see Lines 6-7 on Page 5).

P7, L7: on haze day or non-haze day?

Response:

To be more clear, we have added the description of the region for haze days. As shown in Lines 6-7 on Page 7, we have added the sentence "The haze days are shown in the areas highlighted in grey in Fig. 6.".

P8, L11: I do not see the results of "the spatial transport of pollutants has a significant effect on haze parameters" can be concluded from the above description.

Response:

Thanks for your valuable comment. We have changed the term "spatial transport" into "vertical transport" and added the figure 8 to further describe the vertical transport of pollutants as shown in Fig. R3. Moreover, the descriptions about the vertical transport of pollutants have been added.

"As shown in Fig. 8, the vertical transport of particles could be obtained by comparing hourly variations of PM2.5 mass concentration and Up-Vis at different altitudes in certain period. In Fig. 8 (1), as the PM2.5 mass concentration near the ground decreased, the Up-Vis at the altitude of 0.5 km increased three hours later than that at the altitudes of 0.1 km and 03 km. This indicates pollutants might ascend and prevents the improvement of Up-Vis at the altitude of 0.5 km. In Fig. 8 (2), the Up-Vis at the altitude of 0.5 km increased rapidly, while the Up-Vis at the altitudes of 0.1 km and 0.3 km increased slowly four hours later. This demonstrates the delayed diffusion might result from the descent of pollutants. While the descent of pollutants cause that near-ground PM2.5 mass concentration decreased slowly in this period. Therefore, the delayed variations of Up-Vis between high altitude and low altitude indirectly reveal the influence of vertical transport of pollutants on variation of haze parameters." (see Lines 5-13 on Page 9 and Figure 8 on Page 10).

Finally, the sentence "In addition, the delayed variations of Up-Vis between high altitude

[Figure]

and low altitude reveal the vertical transport of pollutants." has been added to conclude the vertical transport of pollutants (see Lines 16-17 on Page 12).

Figure 7: please give the number of samples.

Response:

We have changed the sentence "As shown in Fig. 7 . . . to describe the effect of spatial transport of particles on haze parameters in the northwest of downtown Beijing." into "As shown in Fig. 7, the correlation between PM2.5 mass concentration and haze parameters was established based on the 201 statistical samples in Fig. 4 and 5, which describes the impact of near-ground particle concentration on haze parameters in the northwest of downtown Beijing." (see Lines 4-6 on Page 8). The sentence "The ABL height is an important parameter to analyze the dynamic effect of air pollution (Wu et al., 2013). HT or AOT directly reflects the pollutant concentrations. The correlations between ABL, HT, AOT, and Up-Vis are plotted in Fig. 8." has changed into "According to the 201 statistical samples mentioned above, the correlations between vertical haze parameters (ABL, HT and AOT) and horizontal haze parameters (Up-Vis) are plotted in Fig. 9 to analyze the two-dimensional characteristic of haze phenomenon." (see Lines 3-5 on Page 10).

P10, L5: please give the standard of the four haze levels

Response:

According to the observation and forecasting levels of haze (QX/T 113-2010) supplied by CMA, when the horizontal visibility on the ground is between 5 km and 10 km, the haze level is slight pollution; when the horizontal visibility on the ground is between 3 km and 5 km, the haze level is mild pollution; when the horizontal visibility on the ground is between 2 km and 3 km, the haze level is moderate pollution; when the horizontal visibility on the ground is less than 2 km, the haze level is severe pollution, which is displayed in table R2. To be more scientific, we have changed the sentence

"According to the observation and forecasting levels of haze supplied by CMA. . ." into "According to the observation and forecasting levels of haze (QX/T 113-2010) supplied by CMA. . ." (see Lines 4-5 on Page 11).

Table R2: Standard of the haze levels. Haze levels Standard/km Slight pollution 5-10 Mild pollution 3-5 Moderate pollution 2-3 Severe pollution <2

References: Chen, H., and Wang, H.: Haze Days in North China and the associated atmospheric circulations based on daily visibility data from 1960 to 2012, Journal of Geophysical Research, 120, 5895-5909, 2015.

Han, R., Wang, S., Shen, W., Wang, J., Wu, K., Ren, Z., and Feng, M.: Spatial and temporal variation of haze in China from 1961 to 2012, Journal of Environmental Sciences, 46, 134-146, 2016.

Tang, G., Zhu, X., Hu, B., Xin, J., Wang, L., Münkel, C., Mao, G., and Wang, Y.: Impact of emission controls on air quality in Beijing during APEC 2014: lidar ceilometer observations, Atmospheric Chemistry and Physics, 15, 12667-12680, 2015.

Please also note the supplement to this comment:
https://www.atmos-chem-phys-discuss.net/acp-2018-30/acp-2018-30-AC2-supplement.pdf

————————————————

**Fig. 1.** Scatter plot of PM2.5 mass concentration and Up-Vis in the northwest of downtown Beijing. The inserted table denotes the statistical gradient of Up-Vis at different altitudes.

**Fig. 2.** Correlation of the AOT values deduced from AERONET sites and ground-based LiDAR data. The inserted chart gives the changes of PM2.5 mass concentration and AOT values at the ground-based LiDAR site on

[Figure]

**Fig. 3.** Hourly variation of Up-Vis and PM2.5 mass concentration in certain period.

---

## Referee Report (RR1)

Comments to the authors:

The scientific meaning and academic value of this article is not described effectively. What is the contribution to the haze study due to this work? The conclusions and abstract are too simple and common, some conclusions are such specious arguments. A considerable part of the references listed is irrelevant to this paper. Authors need to study the relevant study results and articles carefully and refine more meaningful and detailed research goals. English expression need to be refined and modified.

1. "The Up-Vis on non-haze days was about 3-5 times higher than that on haze days." There's a problem with this conclusion.

2. "Moreover, a strong correlation between PM2.5 mass concentration and haze parameters shows an obvious influence of near-ground particle concentration on haze parameters, so the haze phenomenon could be alleviated by controlling pollutant concentrations near the ground". What is the definite scientific meaning if this conclusion?

---

## Referee Report (RR2)

**Review of**

**"Multiplatform analysis of upper air haze visibility in downtown Beijing"**

**General Comments**

This paper presents a comparison between the surface $PM_{2.5}$ mass concentration and upper-air visibility (i.e., visibility at 0.1, 0.3, and 0.5 km) during two winter haze episodes that occurred in the northwest part of downtown Beijing. While use of the term "visibility" for altitudes above the surface may be somewhat misleading, the essence of the study is concerned with a very important but rarely studied issue, namely, the vertical variation of aerosol loading, especially the consistency between ground-level and upper-level measurements monitored by a suite of instruments including the lidar, Cimel sunphotometers, $PM_{2.5}$ particle samplers, etc. Such analyses are useful towards understanding the source of air pollution and its transport, as well as uncertainties in using ground-based measurements to represent column values or vice versa. The study is generally rigorous and sound. In light of these merits, I recommend publication if the following comments are properly addressed.

**Specific comments:**

1. Change the title to "Comparison of Air Quality at Different Altitudes from Multi-Platform Measurements in Beijing".

2. In the abstract, please clarify the altitudes of the upper-air levels under study, the exact research period, and what the haze parameters refer to.

3. None of the AOD observation stations belong to the AERONET whose data is processed and quality-controlled by the NASA AERONET team. The instruments deployed at these stations are the same as AERONET, namely, French-made Cimel sunphotometers. However, the operational mode and retrieval algorithms are not the same as those from the AERONET because different institutions in China are involved.

4. Page 3: The term "data type" is incorrect. Use the term "variables" instead. State the periods of all datasets used here.

5. Page 3: "AOT is classified as vertical haze parameter" is rather misleading because AOT is a column-integrated quantity, i.e., the total loading of aerosols.

6. A description of the algorithm used to retrieve AOD from Raman-Mie lidar signals is needed.

7. The conclusion that "the spatial transport of pollutants has a significant effect on haze parameters" is made. It is unclear how

this conclusion was reached based on the analysis presented.

8. Can you explain why the correlations between surface $PM_{2.5}$ and visibilities at 0.3 km and 0.5 km are much stronger than the correlation between surface $PM_{2.5}$ and visibility at 0.1 km?

---

## Author Response (AR2)

**RESPONSE LETTER**

**Manuscript ID:** acp-2018-30

June 29th, 2018

We deeply appreciate the reviewer for his/her careful reviews of this paper.

**Referee #3. *General Comments**

*This paper presents a comparison between the surface PM2.5 mass concentration and upper-air visibility (i.e., visibility at 0.1, 0.3, and 0.5 km) during two winter haze episodes that occurred in the northwest part of downtown Beijing. While use of the term "visibility" for altitudes above the surface may be somewhat misleading, the essence of the study is concerned with a very important but rarely studied issue, namely, the vertical variation of aerosol loading, especially the consistency between ground-level and upper-level measurements monitored by a suite of instruments including the lidar, Cimel sunphotometers, PM2.5 particle samplers, etc. Such analyses are useful towards understanding the source of air pollution and its transport, as well as uncertainties in using ground-based measurements to represent column values or vice versa. The study is generally rigorous and sound. In light of these merits, I recommend publication if the following comments are properly addressed.*

*Specific comments:*

*1. Change the title to "Comparison of Air Quality at Different Altitudes from Multi-Platform Measurements in Beijing".*

**Reply:**

Thanks for your valuable comment. We have changed the title "Multiplatform analysis of upper-air haze visibility in downtown Beijing" into "Comparison of Air Quality at Different Altitudes from Multi-Platform Measurements in Beijing" (see Lines 1-2 on **Page 1**).

*2. In the abstract, please clarify the altitudes of the upper-air levels under study, the exact research period, and what the haze parameters refer to.*

**Reply:**

We have added the periods of the haze episodes, the altitude of the upper-air visibility and the detailed haze parameters in the Abstract as shown the sentence of "The features of upper-air visibility at altitudes of 0.1km, 0.3km and 0.5km and the two-dimensional haze characteristics in the northwest of downtown Beijing were studied by using a multiplatform analysis during haze episodes between December 17th, 2016 and January 6th, 2017." and the sentence of "In addition, the two-dimensional haze characteristics could be studied by analyzing the correlation between vertical haze parameters (atmospheric boundary layer, haze thickness and aerosol optical thickness) and horizontal haze parameter (upper-air visibility)." (see Lines 7-9 and Lines 12-14 on **Page 1**).

*3. None of the AOD observation stations belong to the AERONET whose data is processed and quality-controlled by the NASA AERONET team. The instruments deployed at these stations are the same as AERONET, namely, French-made Cimel sunphotometers. However, the operational mode and retrieval algorithms are not the same as those from the AERONET because different institutions in China are involved.*

**Reply:**

Thanks for your valuable comment. The AERONET data used in this paper are downloaded from the website of "https://aeronet.gsfc.nasa.gov" (Holben et al., 2001). Using the downloading data of Beijing site, Beijing_RADI site, Beijing_PKU site and Beijing_CAMS site, the aerosol optical thickness (AOT) value in the ground-based LiDAR site can be obtained directly through statistical calculation according to the distance information between the LiDAR site and the selected AERONET sites. Moreover, many researchers have analyzed the global or local AOT variation combining the AERONET data (Chen et al., 2016; Jiang et al., 2007; Kim et al., 2014; Lyapustin et al., 2011; Xiao et al., 2016; Yoon et al., 2016).

Figure R1 shows the AOT variation for the selected four AERONET sits and its statistical horal value for the ground-based LiDAR site in December 31, 2016.

Though the AOT value depends on the location of the AERONET site, the variation trend is basically consistent. Therefore, the calculated statistical horal AOT value for the LiDAR site would reflect sufficiently the actual AOT variation, which is used to demonstrate the feasibility of the retrieved AOT value with the ground-based LiDAR data through comparing each other.

[Figure]

Figure R1: AOT variation for the selected AERONET sites and its statistical value for the ground-based LiDAR site in December 31, 2016.

*4. Page 3: The term "data type" is incorrect. Use the term "variables" instead. State the periods of all datasets used here.*

**Reply:**

We have changed the term "data type" into "variables" as shown the sentence of "The PM2.5 mass concentration is one of the variables to be monitored." (see Line 4 on **Page 3**). To state the periods of all datasets, we have added the sentence "Moreover, the periods of all the downloading PM2.5 mass concentration data and AOT data are the same as the detecting time, between December 13th, 2016 and January 11th, 2017, of ground-based LiDAR site." (see Lines 7-8 on **Page 3**).

*5. Page 3: "AOT is classified as vertical haze parameter" is rather misleading because AOT is a column-integrated quantity, i.e., the total loading of aerosols.*

**Reply:**

Thanks for your valuable comment. The aerosol optical thickness (AOT) is exactly the column-integrated quantity of aerosol extinction coefficient over a certain vertical distance (Dieudonné et al., 2017). It denotes the vertical total loading of aerosols. And because of its representative significance to pollutant concentration at a certain vertical distance, the AOT can be classified as vertical haze parameter.

*6. A description of the algorithm used to retrieve AOD from Raman-Mie lidar signals is needed.*

**Reply:**

Thanks for your valuable comment. Aerosol optical thickness (AOT) can be defined as the extinction of monochromatic light due to the presence of aerosols in the atmosphere. Based on the lidar equation, the aerosol extinction coefficient was retrieved by some robust inversion methods. Then the AOT can be obtained by the integration of aerosol extinction coefficient over a certain vertical distance (Dieudonné et al., 2017). To be more scientific, we have added the sentence "According to the definition of AOT, it can be obtained by the integration of aerosol extinction coefficient over a certain vertical distance with the expression of $\int_0^z \alpha_a(z)dz'$, where $\alpha_a(z)$ is the aerosol extinction coefficient (AEC) which is retrieved from ground-based Raman-Mie LiDAR data with some robust inversion methods (Ji et al., 2017)." to describe the method used to retrieve AOD from Raman-Mie lidar signals (see Lines 20-23 on **Page 3**).

*7. The conclusion that "the spatial transport of pollutants has a significant effect on haze parameters" is made. It is unclear how this conclusion was reached based on the analysis presented.*

**Reply:**

Thanks for your valuable comment. We have added the figure 8 to further describe the vertical transport of pollutants as shown in Fig. R2. Moreover, the descriptions about the vertical transport of pollutants have been added.

"As shown in Fig. 8, the vertical transport of particles could be obtained by comparing hourly variations of PM2.5 mass concentration and Up-Vis at different altitudes in certain period. In Fig. 8 (1), as the PM2.5 mass concentration near the ground decreased, the Up-Vis at the altitude of 0.5 km increased three hours later than that at the altitudes of 0.1 km and 03 km. This indicates pollutants might ascend and prevents the improvement of Up-Vis at the altitude of 0.5 km. In Fig. 8 (2), the Up-Vis at the altitude of 0.5 km increased rapidly, while the Up-Vis at the altitudes of 0.1 km and 0.3 km increased slowly four hours later. This demonstrates the delayed diffusion might result from the descent of pollutants. While the descent of pollutants cause that near-ground PM2.5 mass concentration decreased slowly in this period. Therefore, the delayed variations of Up-Vis between high altitude and low altitude indirectly reveal the influence of vertical transport of pollutants on variation of haze parameters." (see Lines 5-13 on **Page 9** and Figure 8 on **Page 10**).

[Figure]

Figure R2: Hourly variation of Up-Vis and PM2.5 mass concentration in certain period.

Finally, the sentence "In addition, the delayed variations of Up-Vis between high altitude and low altitude reveal the vertical transport of pollutants." has been added to conclude the vertical transport of pollutants (see Lines 17-18 on **Page 12**).

*8. Can you explain why the correlations between surface PM2.5 and visibilities at 0.3 km and 0.5 km are much stronger than the correlation between surface PM2.5 and visibility at 0.1 km?*

**Reply:**

Thanks for your valuable comment.

The surface PM2.5 mass concentration in the ground-based LiDAR site is obtained directly through statistical calculation according to the distance information between the LiDAR site and the selected air quality monitoring sites, including Xizhimen north site, Wanliu site and Guanyuan site. So the statistical PM2.5 mass concentration is more representative for the ground-based LiDAR site.

Considering the location of the detecting sites and the influence of human activities on LiDAR signals at different altitudes, the LiDAR signal at high altitude (at 0.3km and 0.5km) is less affected by human activities, so more accurate and consistent Up-Vis at higher altitude can be obtained. And, Up-Vis at low altitude (0.1km) derived from the LiDAR signal depends on human activities obviously, and shows greater uncertainty.

As shown in Fig. R3, the correlations between surface PM2.5 and Up-Vis at altitudes of 0.3 km and 0.5 km are about 0.81 and 0.76, which are stronger than that at altitude of 0.1 km. To express more clearly, we have added the sentence "Moreover, owing to the location of detecting sites (located at the centre of Beijing) and the different influence of human activities on Up-Vis at individual altitudes, the correlations between surface PM2.5 and Up-Vis at altitudes of 0.3 km and 0.5 km (0.81 and 0.76 respectively) are much stronger than the correlation between surface PM2.5 and Up-Vis at altitude of 0.1 km (0.62)." (see Lines 8-11 on **Page 8**).

[Figure]

Figure R3: Scatter plot of PM2.5 mass concentration and Up-Vis in the northwest of downtown Beijing.

**Referee #1.** *The scientific meaning and academic value of this article is not described effectively. What is the contribution to the haze study due to this work? The conclusions and abstract are too simple and common, some conclusions are such specious arguments. A considerable part of the references listed is irrelevant to this paper. Authors need to study the relevant study results and articles carefully and refine more meaningful and detailed research goals. English expression need to be refined and modified.*

*1.*"*The Up-Vis on non-haze days was about 3-5 times higher than that on haze days.'' There's a problem with this conclusion.*

**Reply:**

Thanks for your valuable comment. The conclusion (The Up-Vis on non-haze days was about 3-5 times higher than that on haze days.) is obtained from the daily variation analysis of multiplatform data between December 13th, 2016 and January 11th, 2017 as shown in Fig. R4. To be more scientific, we have changed the sentence "The Up-Vis on non-haze days was about 3-5 times higher than that on haze days." into "The Up-Vis on non-haze days was about 3-5 times higher than that on haze days with the ground-based Raman-Mie LiDAR data between December 13th, 2016 and January 11th, 2017." (see Lines 11-13 on **Page 12**).

[Figure]

Figure R4: Daily variation of multiplatform data during successive haze episodes in the northwest of downtown Beijing.

*2. "Moreover, a strong correlation between PM2.5 mass concentration and haze parameters shows an obvious influence of near-ground particle concentration on haze parameters, so the haze phenomenon could be alleviated by controlling pollutant concentrations near the ground". What is the definite scientific meaning if this conclusion?*

**Reply:**

Thanks for your valuable comment. As shown in Fig. R5, with the increasing of near-ground PM2.5 mass concentration, the Up-Vis and atmospheric boundary layer (ABL) decrease exponentially, the haze thickness (HT) and aerosol optical thickness (AOT) increase linearly. Therefore, the excellent haze phenomenon can be realized by controlling the fine particle concentration near the ground.

To be more scientific, we have changed the sentence "Moreover, a strong correlation between PM2.5 mass concentration and haze parameters shows an obvious influence of near-ground particle concentration on haze parameters, so the haze phenomenon could be alleviated by controlling pollutant concentrations near the ground." into "Moreover, a strong correlation between PM2.5 mass concentration and haze parameters shows an obvious influence of near-ground fine particle concentration on haze parameters, so the haze phenomenon could be alleviated by controlling fine pollutant concentrations near the ground." (see Lines 15-17 on **Page 12**).

In addition, the essence of this study is concerned with the vertical variation of aerosol loading, especially the consistency between ground-level and upper-level measurements monitored by a suite of instruments including lidar, AERONET, PM2.5 particle samplers, etc. Such analyses not only help to understand the air pollution transport, but also benefit to understand the uncertainties in using ground-based measurements to represent column values. To be more scientific, we have added the sentence "Such analyses are useful to understanding the air pollution transport, as well as uncertainties in using ground-based measurements to represent column values." in the abstract (see Lines 15-16 on **Page 1**).

[Figure]

Figure R5: Scatter plot of PM2.5 mass concentration and haze parameters of Up-Vis, ABL, HT, and AOT in the northwest of downtown Beijing.

**Reference:**

Chen, H., Cheng, T., Gu, X., Li, Z., and Wu, Y.: Characteristics of aerosols over Beijing and Kanpur derived from the AERONET dataset, Atmospheric Pollution Research, 7, 162-169, 2016.

Dieudonné, E., Chazette, P., Marnas, F., Totems, J., and Shang, X.: Raman Lidar Observations of Aerosol Optical Properties in 11 Cities from France to Siberia, Remote Sens., 9, 978, 2017.

Holben, B. N., Tanré, D., Smirnov, A., Eck, T. F., Slutsker, I., Abuhassan, N., Newcomb, W. W., Schafer, J. S., Chatenet, B., and Lavenu, F.: An emerging ground-based aerosol climatology: Aerosol optical depth from AERONET, J. Geophys. Res. Atmos., 106, 12067-12097, 2001.

Jiang, X., Liu, Y., Yu, B., and Jiang, M.: Comparison of MISR aerosol optical thickness with AERONET measurements in Beijing metropolitan area, Remote Sens. Environ., 107, 45-53, 2007.

Kim, M., Kim, J., Man, S. W., Yoon, J., Lee, J., Wu, D., Chan, P. W., Nichol, J. E., Chung, C. Y., and Ou, M. L.: Improvement of aerosol optical depth retrieval over Hong Kong from a geostationary meteorological satellite using critical reflectance with background optical depth correction, Remote Sens. Environ., 142, 176-187, 2014.

Lyapustin, A., Smirnov, A., Holben, B., Chin, M., Streets, D. G., Lu, Z., Kahn, R., Slutsker, I., Laszlo, I., and Kondragunta, S.: Reduction of aerosol absorption in Beijing since 2007 from MODIS and AERONET, Geophysical Research Letters, 38, 415-421, 2011.

Xiao, Q., Zhang, H., Choi, M., Li, S., Kondragunta, S., Kim, J., Holben, B., Levy, R. C., and Liu, Y.: Evaluation of VIIRS, GOCI, and MODIS Collection 6 AOD retrievals against ground sunphotometer observations over East Asia, Atmospheric Chemistry & Physics, 16, 20709-20741, 2016.

Yoon, J., Pozzer, A., Chang, D. Y., Lelieveld, J., Kim, J., Kim, M., Lee, Y. G., Koo, J. H., Lee, J., and Moon, K. J.: Trend estimates of AERONET-observed and model-simulated AOTs between 1993 and 2013, Atmospheric Environment, 125, 33-47, 2016.